# The Global Trend of Drug Resistant Sites in Influenza A Virus Neuraminidase Protein from 2011 to 2020

**DOI:** 10.3390/microorganisms12102056

**Published:** 2024-10-12

**Authors:** Jing Wang, Bei Shen, Lihuan Yue, Huiting Xu, Lingdie Chen, Dan Qian, Wei Dong, Yihong Hu

**Affiliations:** 1CAS Key Laboratory of Molecular Virology & Immunology, Institutional Center for Shared Technologies and Facilities, Pathogen Discovery and Big Data Platform, Shanghai Institute of Immunity and Infection, Chinese Academy of Sciences, Yueyang Road 320, Shanghai 200031, China; wangjing02@sjtu.edu.cn (J.W.); bshen@siii.cas.cn (B.S.); lhyue@siii.cas.cn (L.Y.); ldchen@siii.cas.cn (L.C.); 2University of Chinese Academy of Sciences, Beijing 100045, China; 3Pediatric Department, Nanxiang Branch of Ruijin Hospital, Shanghai 201802, China; tingtingdoctor@sina.com (H.X.); qiandan363@163.com (D.Q.)

**Keywords:** epidemic, trend, influenza A virus, neuraminidase, drug resistant, mutation

## Abstract

Influenza A virus (IAV) causes highly contagious respiratory disease worldwide, so prevention and control of IAV is extremely important. However, overuse of neuraminidase inhibitor (NAI) drugs leads to drug resistance. To explore the up-to-date geographical distribution and evolution of drug-resistant mutations (DRMs) in the NA protein of IAV, 81,492 near full-length NA sequences downloaded from NCBI and GISAID databases, including 34,481 H1N1 and 46,622 H3N2, were processed and analyzed. Our results showed the annual number of NA sequences from 2011 to 2019 continuously increased. Meanwhile, almost 85% of sequences were from developed countries in North America, Europe and Asia. Clustering analysis demonstrated H3N2 varied more than H1N1. Notably, H3N2 exhibited a higher frequency of DRMs than H1N1, with prevailing DRMs mainly located at non-active sites within the NA protein. Phylogenetic analyses showed NA harboring DRMs collected in the same year and from the same location clustered together, which may be related to the local economic level, clinical monitoring of DRMs and research level. Consequently, it is imperative to enhance global surveillance targeting drug resistance in IAV infections which can mitigate the transmission of drug-resistant strains. In summary, our research provides valuable insights for clinical medication while establishing a robust scientific basis for IAV prevention and treatment strategies to improve overall efficacy.

## 1. Introduction

The influenza viruses (IAVs) can infect individuals of all age groups [1]. They infect respiratory epithelial cells and cause cell death in the upper and lower respiratory tract and lung parenchyma [2]. The symptoms are characterized by fever, sore throat, rhinorrhea, cough, cephalalgia, myalgia and fatigue. In addition, IAVs may induce non-respiratory complications that impact the cardiovascular system, central nervous system and other organ systems. Globally, an estimated 500,000 individuals suffer from influenza-related complications annually. And seasonal IAV infection leads to an estimated 3 to 5 million severe cases worldwide every year [3,4].

Neuraminidase (NA), a surface glycoprotein, is an indispensable enzyme for the replication of influenza virus. The polypeptide chain of NA comprises 470 amino acid residues. The monomer NA consists of four domains: the cytoplasmic tail, the transmembrane domain, the globular ‘head’ and the ‘stem’ which connects the head to the transmembrane domain. The enzyme’s active site is composed of conserved amino acids found in all known influenza A and B neuraminidases [5]. Active NA forms a tetramer consisting of four identical subunits, while the monomeric form lacks enzymatic activity. In the absence of NA activity, the virus can undergo a complete cycle of progeny virus production, but it cannot spread to new cells to initiate another round of infection. Therefore, NA plays a crucial role in many stages of the influenza virus infection process. It helps the virus approach target cells by inhibiting sialic acid in proteins, playing an important role in the early stages of viral infection [6]. Additionally, it participates in the fusion process [7]. Finally, the most characteristic function of NA is its role as a sialidase, which involves cleaving sialic acid from carbohydrate side chains on cell surface receptors and newly formed viruses, thereby releasing new viral progeny [8,9].

Administration of antiviral medications is one of the conventional approaches for influenza control. Influenza antiviral drugs are classified into three categories: M2 channel blockers, neuraminidase inhibitors (NAIs) and polymerase inhibitors. As an M2 channel blocker, amantadines were the pioneering antiviral drugs globally employed in influenza treatment. However, due to the high mutation rate, drug resistance against amantadine has escalated significantly over time. Consequently, amantadine has not been recommended as influenza therapy since 2006 [10]. NAIs have emerged as the primary antiviral drugs, also facing the imminent risk of developing resistance.

The enzymatic activity of NA is primarily associated with eight catalytic sites (Arg118, Asp151, Arg152, Arg224, Glu276, Arg292, Arg371, Tyr406) and auxiliary sites at 11 framework residues (Glu119, Arg156, Trp178, Ser179, Asp/Asn198, Ile222, Glu227, His274, Glu277, Asn294 and Glu425) based on N2 numbering. The active sites directly interact with NAIs such as oseltamivir and zanamivir. Mutations in these catalytic sites and framework residues result in drug resistance [11]. The mutations at L134, D243, D330, T365, I117, Q136, G209, N221, V233, S246, D347 and S404 have also been reported to lead to varying degrees of drug resistance [12].

Although significant contributions have been made by antiviral drugs, varied degrees of drug resistance are gradually generated with their increased usage. Therefore, exploring the mechanisms underlying the emergence and spread of antiviral resistance is necessary for effectively controlling both seasonal and pandemic influenza outbreaks. Our study aims to analyze the global trend of NA sequence drug resistance loci in influenza A virus from 2011 to 2020, providing a theoretical foundation for preventing and treating influenza while offering clinical medication guidance.

## 2. Materials and Methods

### 2.1. Sequence Acquisition and Renaming

The NA sequences of all influenza viruses were obtained from NCBI (https://www.ncbi.nlm.nih.gov/genomes/FLU/Database/nph-select.cgi#mainform (accessed on 31 March 2023)) and GISAID (http://gisaid.org/ (accessed on 31 March 2023)). The name formats of the downloaded sequences from the NCBI and GISAID databases are different. To facilitate subsequent analysis, a standardized format is adopted for renaming the sequences based on their accession number, country of origin, continent and year of collection.

### 2.2. Deduplication

Shorter or duplicated sequences were eliminated using the web server provided by NCBI’s Influenza Virus Resource. Python scripts are employed to eliminate redundant sequences in order to complement the shortcomings of NCBI’s processing. Specifically, the Python script was processed by “cmd” command line to execute the designated Python file for deleting duplicates. The Python deldu.exe file is available at github repository (https://github.com/nabeel-pdc (accessed on 31 May 2023)). Subsequently, a total of 81,492 NA sequences of influenza A virus were included for further analysis, comprising 34,481 H1N1 and 46,622 H3N2.

### 2.3. Alignment

Online sequence alignment tool, MAFFT (https://mafft.cbrc.jp/alignment/software/ (accessed on 31 May 2023)), was used for sequence alignment. Subsequently, the aligned sequences were manually trimmed to ensure their qualification as nearly full-length NA sequences. It is important that the name of each aligned sequence should not contain parentheses, which may lead to “sequences with no tips” or “tips with any sequences” which cause interruption in subsequent clustering analysis.

### 2.4. Phylogenetic Analysis

We use FastTree Version 2.1.11 for construction of maximum likelihood phylogenetic trees with treated aligned NA fasta format sequences. Here, the Generalized Time Reversible (GTR) model was used for nucleotide-based datasets. The following command was used, “FastTree -gtr -nt Alignment.file > Tree.file”, and the output file was exported in Newick format for later analysis.

### 2.5. Cluster Analysis

The aligned sequence (fasta file) and the phylogenetic tree (tree Newick file) were imported into ClusterPicker GUI_1.2.3.jar for clustering analysis. Parameters were selected as follows, Initial Threshold = 0.9, Main Support Threshold = 0.9, Genetic Distance Threshold = 4.5 and Large Cluster Threshold = 10. Subsequently, the in-cluster and non-cluster sequences were exported and analyzed.

### 2.6. Analysis of Amino Acid Mutation Sites

The in-cluster and non-cluster consensus sequences were determined using the online Consensus Maker tool (https://www.hiv.lanl.gov/content/sequence/CONSENSUS/SimpCon.html (accessed on 31 May 2023)). MEGA 7.0 software was employed to translate the consensus sequences of both in-cluster and non-cluster sequences into amino acid, enabling comparison of their dissimilarities and identification of DRM sites within the in-cluster amino acid sequences every year.

## 3. Results

### 3.1. Analysis of Influenza A Virus Sequence Numbers Each Year from 2011 to 2020

We analyzed global nearly full-length NA sequences of influenza A virus from 2011 to 2020. These results are shown by year in Figure 1a; it demonstrates a consistent annual increasing number of influenza A NA sequences from 2011 to 2019, with the peak observed in 2019 (16,512). Notably, there was a sharp decline recorded in 2020 (4846). Meanwhile, most of the sequences (85%) come from developed countries (North America 39%, Asia 25%, Europe 21%), as shown in Figure 1b, which may be due to the well-developed local economic, on-time clinical monitoring of DRMs and the research level.

The main IAVs infect humans through the H1N1 and H3N2 strains. After merging and eliminating duplicate sequences retrieved from the NCBI and GISAID databases, a total of 34,481 H1N1 sequences and 46,622 H3N2 sequences from 2011 to 2020 were analyzed (Figure 1c). The number of NA sequences for both subtypes of H1N1 and H3N2 was counted annually. From 2011 to 2019, there was a sustained uptrend for both subtypes. H1N1 reached its peak in 2019 (8129), while H3N2 peaked in 2017 (9048). Compared with H1N1, H3N2 is dominant in the total number, and has become the main epidemic strain. The more people infected by H3N2, the higher the likelihood of mutations and the greater the number of variants that emerge. Additionally, the prevalence analysis of H1N1 and H3N2 across continents reveals that North America, Asia and Europe are the top three continents. Specifically, North America exhibits the highest number of cases for both H1N1 and H3N2, which is nearly equivalent to the combined cases in Asia and Europe. The prevalence of H3N2 and H1N1 in Oceania, Africa and South America remains low, at a maximum of 5%, except for 10% for H3N2 in Oceania (Figure 1d,e). These findings suggest the necessity of enhancing the surveillance of IAV infections globally, especially in non-developed regions, to inhibit IAV transmission.

### 3.2. Geographical Distribution of Influenza A Virus from 2011 to 2020

We comprehensively analyzed the global distribution of IAV infections across seven continents (Figure 2a,b). IAV exhibits predominant prevalence in six continents, North America, South America, Asia, Europe, Oceania and Africa. However, there are no reported cases in Antarctica due to the lack of permanent residents and perpetually low temperatures. In 2011, the prevalence in Asia ranked first, significantly higher than in other continents. Subsequently, the prevalence of IAV in North America continuously increased to the first place. Europe, which had been ranked third, has ranked in second place since 2018. The analysis revealed that IAV sequences in Asia, North America and Europe accounted for over 80% of the annual sequences, indicating a higher prevalence of IAVs. This suggests that economically developed countries across those continents apply more extensive IAV research in terms of sequence data acquisition. Conversely, very limited sequences were reported in Oceania, South America and Africa historically. Despite a general uptrend from 2011 to 2019 across all six continents, IAV infections peaked in 2019 in North America, Europe and Asia, and in previous years in Oceania (2017), South America (2018) and Africa (2018), and a downward trend was witnessed in 2020. This decline might be attributed to the COVID-19 pandemic from the year of 2020, which imposed preventive measures effectively limiting the spread of IAVs.

Similarly, North America, Asia and Europe are the predominant continents for H3N2 (Figure 2e,f) and H1N1 (Figure 2c,d) prevalence. However, there is a difference in the occurrence of peaked years between H3N2 and H1N1. In 2016, H1N1 reached its first peak in North America, Europe, South America and Oceania. Apart from Africa, the remaining five continents experienced their second peak in 2019. H3N2 was prevalent in almost all continents (except South America); two epidemic peaks occurred in 2017 and 2019. From 2012 to 2019, North America was the predominant region consistently for the H3N2 subtype, except 2011. However, there has been a notable increase in H3N2 in Europe since 2016, surpassing Asia and even ranking first in 2020. Additionally, Oceania shows a significantly higher proportion of H3N2 compared with H1N1. Notably, both subtypes demonstrated diverse epidemic trends from 2016 to 2018 (Figure 2c–f), which may be attributed to the local specific epidemic strains in those years.

### 3.3. The Annual Incidence of IAV Clusters between 2011 and 2020

Defined as clusters containing more than two sequences, in-cluster sequences of IAVs from 2011 to 2020 were extracted by ClusterPicker. A total of 1165 clusters were obtained and categorized by year. As shown in Figure 3a, the results reveal a non-linear relationship between the numbers of clusters and sequences over time. Specifically, the number of clusters decreased in 2012, 2014, 2017, 2018 and 2020, while there was an increase in cluster numbers during 2013, 2015, 2016 and 2019, compared to the previous years.

Meanwhile, we counted the H1N1 and H3N2 clusters, resulting in a total of 601 H3N2 and 404 H1N1 clusters. Figure 3b shows that cluster numbers for both IAV subtypes vary from year to year. Specifically, H1N1 showed a downward trend in 2012, 2014, 2016 and 2019, and an upward trend in 2013, 2015, 2017, 2018 and 2020, compared to previous years. Meanwhile, H3N2 exhibited a downward trend in 2012, 2013, 2016, 2018 and 2020 and an upward trend in 2014, 2015, 2017 and 2019. Furthermore, there is a higher transmission cluster number of H3N2 compared with H1N1. H3N2 cluster numbers varied more significantly than H1N1, suggesting greater diversity and higher likelihood of mutation. Notably, H3N2 cluster numbers were less than ten in 2013 and 2018, while H1N1 cluster numbers sharply increased to 68 clusters in 2020. Since the number of transmission clusters varies greatly from year to year, it is necessary to analyze the epidemic trend of IAV transmission clusters regularly every year.

### 3.4. Analysis of DRM Sites from 2011 to 2020

The consensus sequences of NA transmission clusters were computed and translated into amino acid by MEGA, and DRMs were analyzed and are illustrated in Figure 4a. There are DRMs at 19 sites, including R118, E119, D151, D198, H274, T365, Y406 and E425 located in the active site of NA, and the other DRM sites out of this region. A total of 49 mutations were counted in the active sites, and 631 mutations in the non-active sites (Appendix A). Interestingly, there was a notably higher occurrence of DRMs at non-active sites in comparison to active sites. In the active site of NA, the amino acid substitutions are single or multiple kinds of amino acids. It is impressive that D151 stood out with three replacements by G, ten replacements by N and one replacement by E, while other DRM sites had only one kind of substituted amino acid. There are more sequences harboring diversified mutations at the non-active site, with distinct mutations at I117, G209, N221, V233, D347 and S404. The prevalent DRM is the N221 site, substituted mostly by D (520), with only a few by K and one by S. Although the DRM frequency is higher in non-active sites, their influence on drug resistance to NAIs is very limited.

Our decent analysis of all IAV NA sequences, clusters and clusters harboring DRMs are clearly summarized in Table 1. The number and percentage of IAV sequences in different geographical distribution are listed by year. The top three percentages are highlighted. We conducted a similar analysis for NA clusters and clusters harboring DRMs, and the important drug-resistant IAV epidemic years are spotlighted. Furthermore, an in-depth analysis of DRMs in both H1N1 and H3N2 subtypes was explored subsequently, and the results are summarized and shown.

Then, the DRMs in the consensus sequences of H1N1 and H3N2 were analyzed, respectively, and the results are shown in Figure 4b,c. It has been observed that H1N1 harbors varied numbers of mutations at positions H274, I117, G209, N221, V233 and S404 (Appendix A). Notably, the DRM at site H274Y associated with oseltamivir appeared in seven distinct clusters in 2011, 2012, 2014, 2015 and 2020. Meanwhile, there is a significantly higher prevalence of mutations in the non-active region compared with the active center in NA of H1N1. H3N2 mutations occur at drug-resistant sites 119, 151, 222, 365, 371, 406 and 425 as well as at positions 155, 221, 347 and 404 with varied degrees (Appendix A). Like H1N1, the incidence of DRMs in the non-active sites of H3N2 is greater than that in the active sites, and this trend is more obvious than in H1N1, which has more DRMs. The results show H274Y mutation exclusively occurred within the H1N1 subtype (Appendix A). However, no other mutations were found in the active center of H1N1, while mutations including I117M, G209D, V233L and S404T were identified in framework residues. On the other hand, mutations detected only in H3N2 were E119V, D151G/N, I222V, T365I, R371L, Y406S, E425L, Y155F, D347G and S404G (Appendix A). Interestingly, T365I, R371L, Y406S and E425L were found in 2015 and 2016, and disappeared in later years. Both N221 and S404 sites were found mutated simultaneously in both subtypes. In residue 221, both subtypes harbored a mutation from N to D or K, while H1N1 showed the S404T mutation at residue 404 and H3N2 displayed the S404G mutation. Additionally, annual mutations in H3N2 exclusively occurred at position N221, with no observed constant annual mutations in any residue site of H1N1.

The number of consensus sequences harboring mutations each year is quantified in Appendix A. The lined graphs (Figure 4d) show that H1N1 had a relatively low mutation rate, with no more than 10 mutations produced from 2011 to 2019. However, there was a sudden surge in the number of mutations in 2020, reaching a peak of up to 48 mutations, with an increase of approximately 8 to 48 times compared to previous years. Notably, K substitutions were predominant at the N221 site among these mutations. Meanwhile, the H3N2 virus had a higher frequency of mutations, with only 3 mutations observed from 2011 to 2013 (Figure 4e). However, there was a significant surge in mutation rates from 2014 to 2017, resulting in an abrupt 16 to 25 times increase to 50 mutations in 2014. Subsequently, the mutation count reached its peak at 155 in 2017, primarily attributed to the N221 site mutation.

### 3.5. Phylogenetic Characterization of the Drug-Resistant Strains

To elucidate the evolutionary relationships of IAV NA drug-resistant strains, we construct a phylogenetic tree (Figure 5a,b) with clusters harboring DRMs. As the results show, the sequences clustered together originated in the same year or close years, indicating they have a close evolutionary relationship. It is noteworthy that the 2015-Clust74.fasta and the reference sequence A/California/07/2009/H1N1 cluster together with a closest genetic distance. Furthermore, the drug-resistant viruses are all closely associated with A/California/07/2009/H1N1 and may evolve from it. Moreover, a potential evolutionary relationship is also observed between 2015-Clust74 and its neighbored clusters including 2011-Clust4.fasta, 2012-Clust4.fasta, 2012-Clust12.fasta, 2012-Clust13.fasta, 2013-Clust19.fasta, 2014-Clust32.fasta and 2016-Clust41.fasta. In the case of H3N2, these drug-resistant clusters, 2013-Clust17.fasta and 2020-Clust17.fasta have emerged together with A/New York/392/2004/H3N2 and neighbored clusters, 2011-Clust98.fasta, 2011-Clust57.fasta and 2012-Clust75.fasta, indicating their evolutionary kinship.

Subsequently, we focused on the DRMs in active sites and constructed an evolutionary phylogenetic tree with all sequences of both H1N1 and H3N2 subtypes harboring the mutations (Figure 5c,d). It revealed that all H1N1 sequences with DRMs were from Asia, North America and Europe, while only one sequence was from Oceania. Similarly, all mutated H3N2 sequences were identified in Asia, the Americas and Oceania. Additionally, a higher degree of genetic diversity was found in drug-resistant strains of H3N2 compared with H1N1, resulting in more distinct transmission clusters. Furthermore, our analysis of both subtypes revealed a strong similarity and tendency for clusters originating in the same continents or regions, exhibiting a remarkably close evolutionary relationship. Notably, clust26_1735135_Australia_Oceania_2020/2/2 forms a distinct branch with clusters originating in Asia in 2014, implying a potential evolutionary link between the Oceania strain in 2020 and its predecessors in 2014.

## 4. Discussion

We counted the nearly full-length NA sequences of IAVs from 2011 to 2020 and observed a progressive annual escalation from 2011 to 2019 before 2020. The sudden decreased number of IAV NA sequence in 2020 may due to the COVID-19 outbreak along with community-wide restrictions on activities as well as protective measures, such as wearing masks while venturing outdoors and practicing frequent hand hygiene. The global pandemic prevention and control measures are likely to affect IAV spread and reduce the incidence of influenza as well as hospitalizations and mortality from 2020 to 2021. Furthermore, regional imbalance in the number of IAV NA sequences suggests we strengthen surveillance in researching drug resistance, while focusing on non-developed countries and regions can provide more robust support for IAV prevention and control.

Notably, there are predominately more H3N2 sequences than H1N1, which is consistent with previous investigations that there has been a higher frequency of H3N2 epidemics compared with influenza A/H1N1 and B viruses since the H3N2 virus started circulating in humans in 1968 [13]. This has led to higher morbidity and mortality during the epidemic season, posing a significant health challenge [13]. During the high prevalent season of influenza, H3N2 emerges as the predominant subtype among all influenza subtypes and exhibits the highest mortality [13]. Interestingly, the dominance of H3N2 waned significantly worldwide in 2020. Furthermore, there existed notable disparities between the incidences of H3N2 and H1N1. Based on surveillance data from the US Centers for Disease Control and Prevention (CDC), a relatively substantial number of H1N1 infections were reported in 2020 [14,15], followed by an epidemic peak in 2023 spring and winter [15,16]. This implies an emergence of H1N1 as the predominant strain in future influenza seasons, potentially surpassing H3N2. Meanwhile, based on the latest ‘Influenza Surveillance Weekly’ monitoring data released on 23 February 2023 by the National Influenza Center in China, there is a continuous upward trend in the H1N1 positive rate at 71% of all detected IAVs in China [17]. It appears that H1N1 is poised to play a significant role in the 2023 seasonal influenza outbreak, especially in China. However, H3N2 became the prevalent subtype in 2023 winter in China, surpassing H1N1. Therefore, continuous monitoring of influenza infection and data is imperative.

Overall, there are more clusters in IAV than IBV [18]. Interestingly, we observed that the number of IAV clusters did not exhibit a linear increase over time but rather displayed an alternating pattern, different from that observed in IBV, which demonstrates an increased trend year by year [18]. Moreover, the annual number of clusters for both subtypes of IAVs exhibited distinct patterns, with H3N2 demonstrating a more pronounced variation, which supports that H3N2 is prone to mutate with a higher level of diversity. Compared with H1N1 and IBVs, H3N2 exhibits a faster antigenic evolution with higher epidemic potential and more frequent antigenic drift [19], thereby increasing the frequency of mutations. Our investigation revealed a greater number of DRMs in IAV compared with previous analysis in IBV with only mutation at D197 [18].

The primary association of oseltamivir resistance in IAV is observed with the H274Y mutation, resulting in a significant 1500-fold decrease in susceptibility to oseltamivir compared with wild-type viruses. Additionally, this mutation leads to a 500-fold reduction in sensitivity to peramivir. However, it does not impact zanamivir sensitivity [20]. The H274Y mutation is more likely to arise in response to the selective pressure of oseltamivir on N1 neuraminidase [21]; once H274 has mutated, the effectiveness of oseltamivir is significantly diminished, suggesting the H274Y mutation in an active center is a dominant oseltamivir-resistant mutation, which is consistent with our findings that exclusively detected H274Y mutation within the H1N1 subtype. Meanwhile, multiple mutations within the H3N2 subtype, including E119V, D151G/N and I222V, were identified. Moreover, rare mutations T365I, R371L and Y406S were also identified from 2016 to 2017 in H3N2 subtype. Additionally, E425L, a novel mutation, was observed for the first time. Although numerous mutations were analyzed, the major mutations were located in non-active regions with negligible impact on drug sensitivity. The findings are consistent with the resistance-surveillance reports from the National Influenza Center in China, which indicates that all H3N2 subtypes of IAV and IBV remained susceptible to NAIs and polymerase inhibitors until 1 April 2024 [22].

Phylogenetic trees constructed with the consensus sequences harboring DRMs show that sequences from the same or similar years exhibit a higher tendency to cluster together, suggesting a close relationship. Additionally, there are instances where earlier year sequences are included, indicating the evolution of later-year strains from their predecessors. Moreover, sequences within the same geographical region tended to form distinct clusters, potentially attributed to an increased likelihood of transmission within or between adjacent regions, which is consistent with previous research [23]. Furthermore, consensus sequences from distinct regions also form discrete clusters, potentially influenced by factors such as migratory bird movements [24], human travel patterns and other contributing factors [25].

Here, the limitations of this work are summarized. Firstly, we employed clustering methods and exclusively analyzed clusters comprising more than two sequences, although all the NA near full-length sequences of the IAV from 2011 to 2020 were acquired. So it is not guaranteed that DRMs do exist within these not-selected clusters. Secondly, due to the relatively higher availability of data in North America, Europe and Asia compared with Africa, Oceania and South America, it remains uncertain whether all sequences have been included. The uncertainty may potentially result in an underestimation of the prevalence of DRMs. Third, we examined a total of 8 catalytic sites, 11 auxiliary sites and several previously reported amino acid mutations in NA protein. However, it should be noted that our screening did not encompass all amino acids, potentially leading to the omission of novel mutations.

In future clinical treatment, there is an inevitable trend to specify a personalized therapy plan according to the drug-resistance status of each influenza infected patient. We are going to use the clinical samples collected for genomic sequencing and new method design to detect DRMs of IAVs in our future studies, which is economic, convenient and fast, to evaluate the drug-resistance level. We believe this can provide valuable reference for treatment strategy development of IAV infections and future research and prevent IAV prevalence.

## 5. Conclusions

In conclusion, this is the first comprehensive analysis of global NAI resistance mutations in IAV NA sequences over the past decade. The results demonstrate that H3N2 exhibits a higher prevalence of DRMs compared with H1N1, with H1N1 commonly acquiring the H274Y mutation, while H3N2 develops E119V, D151G/N and I222V mutations. Viral strains from the same year or region tend to exhibit clustering on adjacent branches, indicating that geographical proximity may play a pivotal role in the spread of viruses, which could potentially be attributed to bird migration or increased air travel. Notably, mutations such as T365I, R371L, Y406S and E425L were identified among H3N2 subtypes during the 2016–2017 season and formed transmission clusters. However, most mutations were detected in the non-active region and exhibited limited impact on IAV’s drug sensitivity which implies that most strains retain their susceptibility to anti-influenza drugs, thereby affirming the continued efficacy of current antiviral therapy for IAV. Our research holds significant implications for guiding clinical medication, supporting influenza prevention and treatment efforts and enhancing our capacity to prevent and control influenza outbreaks.

## Figures and Tables

**Figure 1 microorganisms-12-02056-f001:**
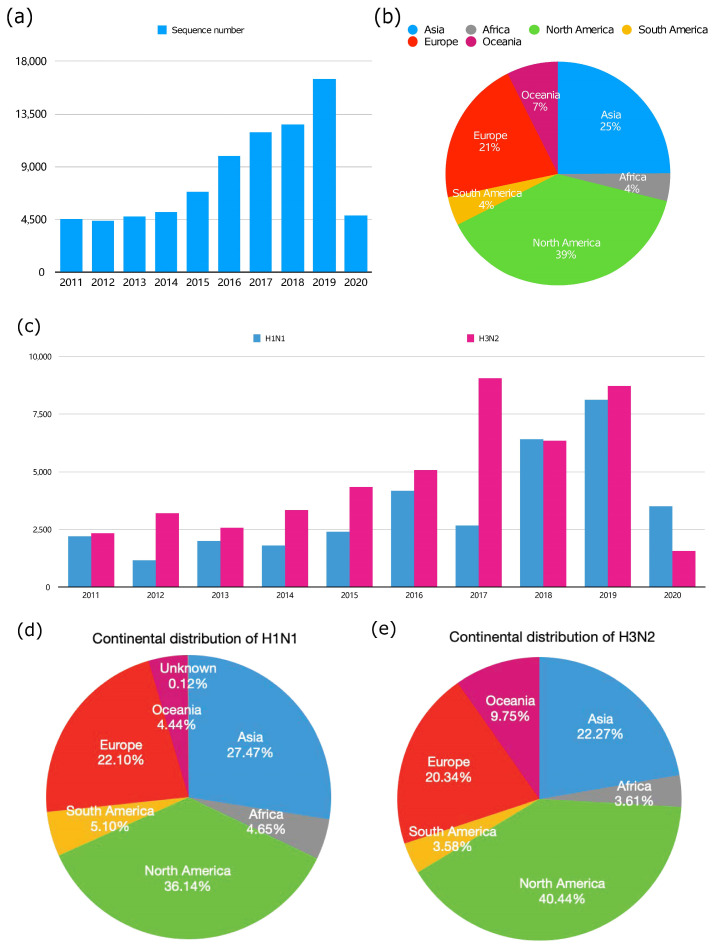
Statistical analysis of influenza A virus sequences from 2011 to 2020. (**a**) Annual sequence numbers of IAV from 2011 to 2020. (**b**) Distribution of the total IAV sequences in each continent from 2011 to 2020. The sequences were respectively downloaded from the NCBI and GISAID databases, deduplicated and counted. (**c**) The number of sequences of H1N1 and H3N2 in each year from 2011 to 2020. (**d**) Geographical distribution of total sequence number of H1N1 from 2011 to 2020 by continent. (**e**) Geographical distribution of total sequence number of H3N2 from 2011 to 2020 by continent.

**Figure 2 microorganisms-12-02056-f002:**
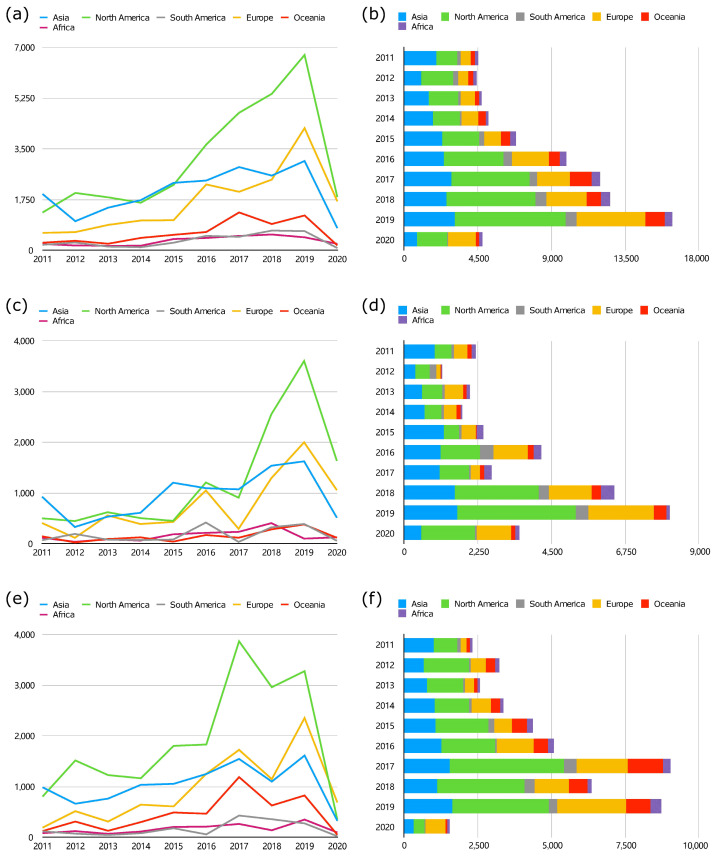
Distribution of the IAV sequences by continent for each year from 2011 to 2020. (**a**,**b**) The distribution of each continent’s total annual sequence number of IAV. (**c**,**d**) The distribution of each continent’s annual sequence number of H1N1 subtype. (**e**,**f**) The distribution of the annual sequence number of H3N2 subtype in each continent.

**Figure 3 microorganisms-12-02056-f003:**
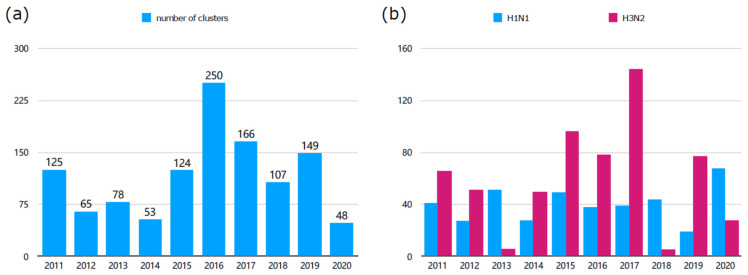
The cluster number of influenza A virus sequences from 2011 to 2020. (**a**) The number of IAV clusters in each year from 2011 to 2020. (**b**) Comparison of H1N1 and H3N2 cluster numbers each year from 2011 to 2020. The rose red bar represents H3N2 subtype, and the blue bar represents H1N1 subtype. The sequences downloaded from NCBI and GISAID databases were merged, deduplicated and aligned with MAFFT, built with FastTree and clustered with ClusterPicker. The number of qualified clusters was found and is shown.

**Figure 4 microorganisms-12-02056-f004:**
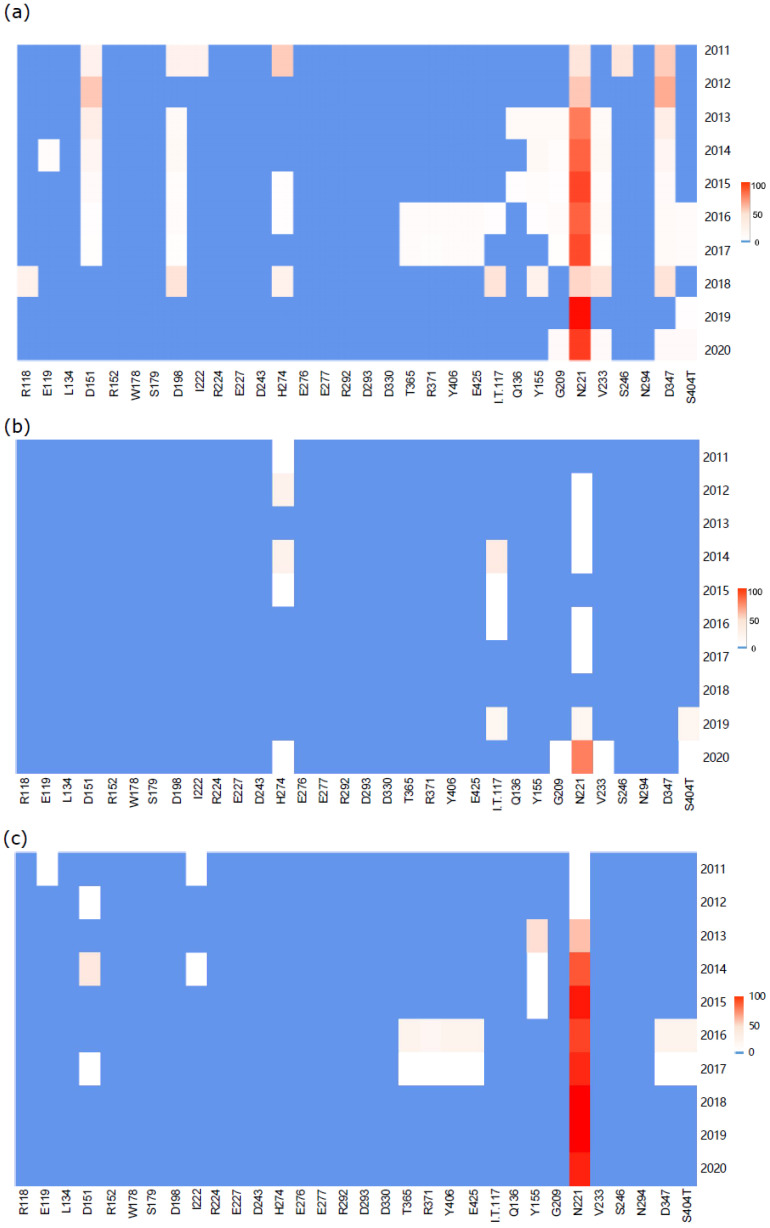
Drug-resistant mutation analysis in the consensus sequences of NA transmission clusters from 2011 to 2020. (**a**) Frequency of the DRMs in consensus sequences at IAV-related sites in each year from 2011 to 2020. (**b**) DRM frequency in consensus sequences at H1N1-associated loci each year from 2011 to 2020. (**c**) DRM frequency in consensus sequences at H3N2-related sites each year from 2011 to 2020. After the consensus sequences were translated into amino acids, the drug-resistant mutations were counted. The values 0–100 represent the percentage. (**d**,**e**) The total number of consensus sequences harboring DRMs in H1N1 and H3N2.

**Figure 5 microorganisms-12-02056-f005:**
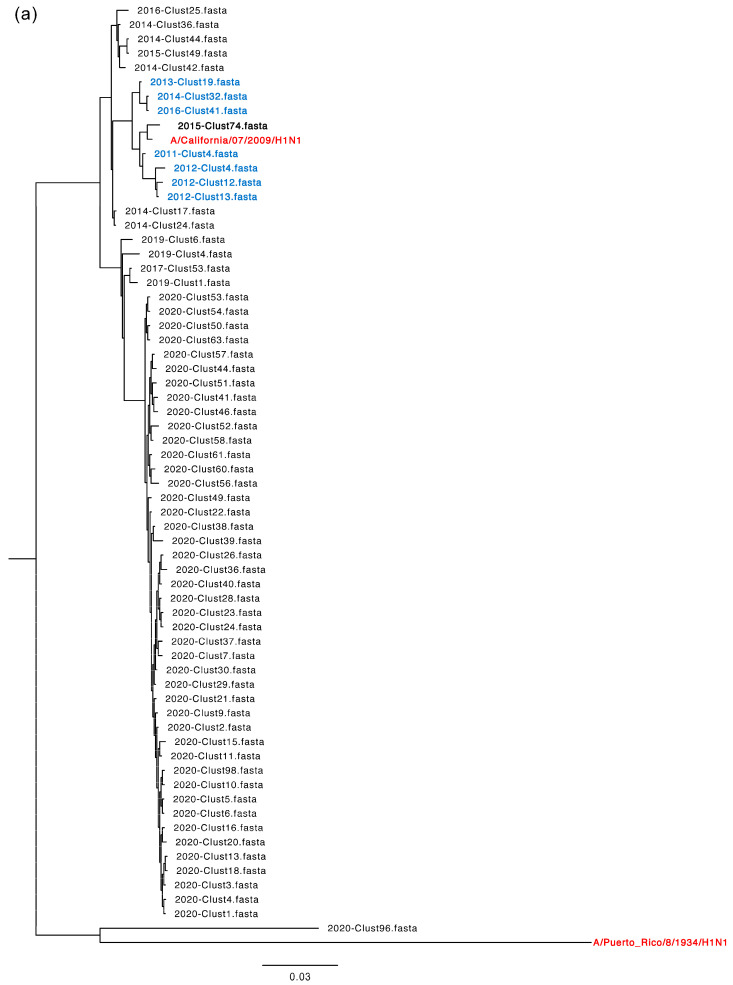
Evolutionary relationships of IAV NA drug-resistant strains from clusters harboring DRMs. (**a**) Phylogenetic tree of H1N1 drug-resistant strains formed clusters. (**b**) Phylogenetic tree of H3N2 drug-resistant strains formed clusters. The bold black indicates the most important DRM-harboring clusters, the bold blue indicates their neighbored clusters, the bold red indicates reference strains. (**c**) Phylogenetic tree of H1N1 NA sequences with DRMs in clusters. (**d**) Phylogenetic tree of H3N2 NA sequences with DRMs in clusters. Red, blue, green and purple represent North America, Asia, Europe and Oceania, respectively.

**Table 1 microorganisms-12-02056-t001:** The global trend of drug resistant sites in influenza A virus neuraminidase protein from 2011 to 2020.

Year	All IAV Sequences (Number/Percentage)														All IAV Clusters %		All IAV Clusters harboring DRMs %	
	Worldwide %		NA ^1^ %		SA ^1^ %		AS ^1^ %		EUR ^1^ %		OCE ^1^ %		AFR ^1^ %					
2011	4542	5.59	1309	4.17	194	5.74	1947	9.62	605	3.59	267	4.42	220	6.74	125	10.73	15	2.44
2012	4401	5.42	1984	6.31	272	8.04	1009	4.99	633	3.75	332	5.50	171	5.24	65	5.58	7	1.14
2013	4711	5.80	1834	5.84	132	3.90	1475	7.29	880	5.22	230	3.81	160	4.90	78	6.70	28	4.55
2014	5124	6.31	1648	5.24	111	3.28	1737	8.58	1033	6.13	432	7.15	163	4.99	53	4.55	51	8.28
2015	6837	8.42	2262	7.20	267	7.89	2334	11.53	1043	6.19	538	8.91	393	12.03	124	10.64	116	** 18.83 **
2016	9917	12.21	3659	11.64	501	**14.81**	2413	11.92	2278	**13.51**	636	10.53	430	13.17	250	**21.46**	89	14.45
2017	11,931	**14.69**	4746	**15.10**	471	13.93	2878	**14.22**	2025	12.01	1311	**21.71**	500	**15.31**	166	**14.25**	138	** 22.40 **
2018	12,577	**15.49**	5401	**17.19**	685	**20.25**	2584	**12.77**	2446	**14.51**	912	**15.10**	549	**16.81**	107	9.18	8	1.30
2019	16,376	**20.17**	6742	**21.45**	667	**19.72**	3088	**15.26**	4219	**25.03**	1207	**19.98**	453	**13.87**	149	**12.79**	124	** 20.13 **
2020	4793	5.90	1843	5.86	82	2.42	770	3.81	1696	10.06	175	2.90	227	6.95	48	4.12	40	6.49
All	81,209	100.00	31,428	100.00	3382	100.00	20,235	100.00	16,858	100.00	6040	100.00	3266	100.00	1165	100.00	616	100.00
**Year**	**H1N1 sequences (number/percentage)**														**H1N1** **clusters %**		**H1N1 clusters harboring DRMs %**	
	**Worldwide %**		**NA %**		**SA %**		**AS %**		**EUR %**		**OCE %**		**AFR %**					
2011	2186	6.35	505	4.05	73	4.15	928	9.80	411	5.39	149	9.74	120	7.49	41	10.15	1	1.59
2012	1170	3.40	450	3.61	197	11.21	333	3.52	120	1.57	28	1.83	42	2.62	27	6.68	3	** 4.76 **
2013	1992	5.78	625	5.02	83	4.72	537	5.67	562	7.37	96	6.27	89	5.56	51	**12.62**	1	1.59
2014	1785	5.18	509	4.09	78	4.44	613	6.47	392	5.14	128	8.37	65	4.06	28	6.93	6	** 9.52 **
2015	2410	7.00	454	3.64	85	4.84	1205	**12.72**	433	5.68	44	2.88	189	11.80	49	**12.13**	2	3.17
2016	4171	**12.11**	1209	9.70	421	**23.96**	1097	11.58	1051	13.79	175	**11.44**	218	**13.61**	38	9.41	2	3.17
2017	2678	7.78	910	7.30	39	2.22	1074	11.34	297	3.90	121	7.91	237	**14.79**	39	9.65	1	1.59
2018	6414	**18.62**	2557	**20.52**	327	**18.61**	1540	**16.26**	1295	**16.99**	286	**18.69**	409	**25.53**	44	10.89	0	0.00
2019	8114	**23.56**	3605	**28.93**	393	**22.37**	1627	**17.18**	2002	**26.27**	383	**25.03**	104	6.49	19	4.70	3	** 4.76 **
2020	3521	10.22	1636	**13.13**	61	3.47	517	5.46	1058	**13.88**	120	7.84	129	8.05	68	**16.83**	44	** 69.84 **
All	34,441	100.00	12,460	100.00	1757	100.00	9471	100.00	7621	100.00	1530	100.00	1602	100.00	404	100.00	63	100.00
**Year**	**H3N2 sequences (number/percentage)**														**H3N2** **clusters %**		**H3N2 clusters harboring DRMs %**	
	**Worldwide %**		**NA %**		**SA %**		**AS %**		**EUR %**		**OCE %**		**AFR %**					
2011	2350	5.00	807	4.29	121	7.26	990	9.54	194	2.05	126	2.77	86	5.116	66	10.98	3	0.64
2012	3250	6.92	1520	8.07	75	4.50	666	6.42	519	5.48	314	6.91	125	7.436	51	8.49	3	0.64
2013	2587	5.51	1231	6.54	48	2.88	767	7.39	314	3.32	132	2.91	72	4.283	6	1.00	1	0.21
2014	3393	7.22	1168	6.20	84	5.04	1039	10.02	647	6.83	303	6.67	117	6.96	50	8.32	44	9.38
2015	4407	9.38	1805	9.58	182	10.92	1058	10.20	613	6.47	493	10.85	208	**12.37**	96	**15.97**	95	** 20.26 **
2016	5135	10.93	1833	9.73	61	3.66	1253	**12.08**	1257	**13.27**	469	10.33	213	**12.67**	78	**12.98**	74	15.78
2017	9144	**19.47**	3869	**20.54**	433	**25.97**	1550	**14.94**	1728	**18.24**	1191	**26.22**	267	**15.88**	144	**23.96**	140	** 29.85 **
2018	6425	**13.68**	2963	**15.73**	360	**21.60**	1103	10.63	1152	12.16	631	**13.89**	142	8.447	5	0.83	5	1.07
2019	8806	**18.75**	3280	**17.42**	278	**16.68**	1617	**15.59**	2356	**24.87**	829	**18.25**	353	21	77	12.81	77	** 16.42 **
2020	1554	3.31	356	1.89	25	1.50	329	3.17	692	7.31	54	1.19	98	5.83	28	4.66	27	5.76
All	47,051	100.2	18,832	100	1667	100	10,372	100	9472	100	4542	100	1681	100	601	100	469	100

^1^ North America (NA), South America (SA), Asia (AS), Europe (EUR), Oceania (OCE), Africa (AFR).

## Data Availability

Data used in the study is available at NCBI Genbank and GISAID. Further, renamed and deduplicated datasets and their metadata will be provided upon request.

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
