# Peer review of "The Global Trend of Drug Resistant Sites in Influenza A Virus Neuraminidase Protein from 2011 to 2020"

_microorganisms, 2024, doi:10.3390/microorganisms12102056_

Round 1
Reviewer 1 Report
Comments and Suggestions for Authors
The manuscript microorganisms-3192043, by Wang and colleagues, provides a comprehensive analysis of the evolution of drug resistance in Influenza A virus neuraminidase over the past decade. Although the study is well-structured, utilizing a robust dataset and appropriate bioinformatics tools to identify significant trends, some points requires revision and are listed below in order of occurrence in the text:
- Line 28: The keywords "Epidemic trend" and "Drug resistance mutation" require revision to match the Medical Subject Headings (MeSH).
- Lines 41-42/124-125: The sentences "The genes encoding the active sites, exhibiting high conservation in both influenza A and B viruses, which cave in the viral 'head'" and "Moreover, as a greater number of individuals being infected, resulting in a higher mutation frequency of H3N2" require revision to make sense.
- Line 52: Insert "(NAIs)" after "neuraminidase inhibitors" as this achronym is used in the rest of the text.
- Line 121: According to Figure 1c, the sustained uptrend for H1N1 and H3N2 was observed from 2011 to 2019, not from 2011 to 2020 as stated.
- Line 155: The sentence "IAV infections peaked in 2019" is only true for North America, Europe and Asia, as such infections peaked in previous years in Oceania (2017), South America (2018) and Africa (2018) according to Figure 2a.
- Lines 156-157: Since the COVID-19 outbreak was declared as a pandemic by the World Health Organization (WHO) only in 11 March 2020, the statement "This decline might be attributed to the COVID-19 pandemic from the end of year 2019" is not accurate.
- Lines 184-185/189-190: Statistical analyses (i.e., the results of hypothesis testing) revealing that cluster numbers for both IAV subtypes varied from year to year and that H3N2 clust numbers varied more significantly than H1N1 are not explicit in Figure 3b-d.
- Line 195: Considering the data shown in Figure 3b, Figure 3c-d is redundant and should be removed.
- Lines 205-212: Previous data on catalytic/auxiliary sites associated with the enzymatic activity of NA as well as mutations in these residues resuting in drug resistance are best suited for the introduction section.
- Line 215: The residue position 365 in NA was not indicated as a hotspot for drug resistance on lines 205-212.
- Lines 231-233: The sentence "Once H274 mutated, the effectiveness of oseltamivir is significantly diminished, suggesting the H274Y mutation in active center is a dominant oseltamivir resistant mutation" is best suited for the discussion section.
- Line 261: The indications "matrix 4", "matrix 1" and "matrix 3" should be removed from Figure 4a-c; in addition, the color gradients in this same figure require revision to better match the corresponding matrices.
- Lines 239/243: In the supplementary information, the number of amino acid variations in H1N1 and H3N2 is shown in Supplementary Tables 3 and 2, respectively (not in the reverse order as stated).
- Lines 256-258: According to Supplementary Table 4, 3 (not 2 to 3 as stated) mutations were observed in H3N2 from 2011 to 2013, thus Figure 4e requires revision to correctly represent this data; in addition, this supplementary table shows that there was a significant surge in mutation rates for H3N2 from 2014 to 2017 (not 2014 to 2016 as stated).
- Lines 272-281: The DRM-harboring clusters indicated in the main text should be highlighted in Figure 5a-b to facilitate their identification.
- Line 288: The sentence "all mutated H3N2 sequences were identified in Asia, North America, and Oceania" is not accurate, as Figure 5d shows two sequences from South America, namely Clust59_543553_Brazil_South_America_2014/3/17 and Clust59_543560_Brazil_South_America_2014/3/10.
- Line 374: The sentence "we examined a total of 9 catalytic sites" is not consistent with the statement on line 205, where it is informed that "The enzymatic activity of NA is primarily associated with eight catalytic sites".
Comments on the Quality of English LanguageModerate editing of English language required.
Reviewer 2 Report
Comments and Suggestions for Authors
This review manuscript has scientific merit that might benefit readers, but some major revisions still need to be included. Therefore, we recommend that the authors make the following modifications:
- Line 22-25: “Phylogenetic analyses showed NA harboring DRMs collected in the same year and from the same location clustered together. Consequently, it is imperative to enhance global surveillance targeting drug resistance in IAV infections, which can mitigate the transmission of drug-resistant strains.”
The authors are suggested to add clinical quantitative data from their study in the above-mentioned abstract part. This addition will make the abstract more comprehensive.
- Line 29-30: “The influenza viruses (IAVs) could infect individuals of all age groups.”
The authors are suggested to add a one-line statement about IAVs that discusses their nature, including how the body responds to IAV infections, such as IAVs, latent infections (e.g., herpes simplex virus), and chronic infections (e.g., hepatitis B/C). This will help the reader understand the pathological response of IAVs, allowing for a better clinical perspective.
- Line 44-45: “Previous data suggests that influenza viruses can complete an entire cycle of viral replication even in the absence of NA activity.”
The authors are suggested to add a reference to this statement and also explain and justify how IAVs complete their cycle without NA. It would be beneficial to create a brief table and include a literature survey about that claim.
Line 45-47: “However, they are unable to effectively spread and initiate further infection. Therefore, NA plays a crucial role in facilitating influenza virus infection, including but not limited to targeting host cells.”
The authors claim in the above lines 44-45 that IAVs can complete their replication in the absence of NA, but the statement above suggests that NA plays a crucial role in facilitating IAV infection, including but not limited to targeting host cells. Please explain this statement and revise it if it is not scientifically proven.
- Section RESULTS 3.2: Geographical distribution of influenza A virus from 2011 to 2020
The authors are suggested to create a table that will demonstrate all the findings from the research and highlight the most prominent work, providing it with more spotlight in a descriptive and brief manner.
Table 1. The Global Trend of Drug Resistant Sites in Influenza A Virus Neuraminidase Protein from 2011 to 2020
Continents |
Predominant prevalence of IAVs |
Annually increasing ratio of IAVs |
Mutants % produced from 2011 to 2019 |
Asia |
|||
Africa |
|||
Europe |
|||
North America |
|||
South America |
|||
Oceania |
- Line 373-377: “The uncertainty may potentially result in an underestimation of the prevalence of DRMs. Third, we examined a total of 9 catalytic sites, 11 auxiliary sites, and several previously reported amino acid mutations in the NA protein. However, it should be noted that our screening did not encompass all amino acids, potentially leading to the omission of novel mutations.”
The authors did a good job in mentioning the limitations of the study. We suggest that the authors add future directions for the study, focusing on the global aspects of IAVs in a more comprehensive way. Kindly add a brief explanation about this so that researchers can follow this issue for future research.

Round 2
Reviewer 1 Report
Comments and Suggestions for Authors
The authors of the manuscript microorganisms-3192043 have addressed most of the criticisms and suggestions raised in my previous review, leading to improvements in the text. However, minor issues still remain to be addressed, as follows:
- Regarding my previous comment "previous data on catalytic/auxiliary sites associated with the enzymatic activity of NA as well as mutations in these residues resuting in drug resistance are best suited for the introduction section", the authors have moved the related text to the materials and methods section, not to the introduction section as suggested.
- Regarding my previous comment "the DRM-harboring clusters indicated in the main text should be highlighted in Figure 5a-b to facilitate their identification", the authors should provide the meaning of the highlight colors in the figure legend.
- Regarding the new table (Table 1) and its related text, the column "H1N1 clusters haboring DRMs %" has only 2 top percentages highlighted, not three as indicated; moreover, its is not clear what the authors meant by the phrase "Furthermore, the in-depth analysis of DRMs in both H1N1 and H3N2 subtypes must be done".
Comments on the Quality of English LanguageModerate editing of English language required.
Reviewer 2 Report
Comments and Suggestions for Authors
The authors have addressed all the comments and suggestions and made the necessary revisions, making the manuscript more comprehensive and scientifically sound. We wish the authors good luck.
Kind wishes.
Author Response
Thanks for the reviewers conclusion that we have addressed all the comments and suggestions.